# Dopamine, Psychosis, and Symptom Fluctuation: A Narrative Review

**DOI:** 10.3390/healthcare10091713

**Published:** 2022-09-07

**Authors:** Gabriela Novak, Mary V. Seeman

**Affiliations:** 1The Integrative Cell Signalling Group, Luxembourg Centre for Systems Biomedicine (LCSB), University of Luxembourg, Esch-sur-Alzette, 4362 Luxembourg, Luxembourg; 2Department of Psychiatry, University of Toronto, Suite #605, 260 Heath St. West, Toronto, ON M5P 3L6, Canada

**Keywords:** schizophrenia, neurodevelopment, oligodendrocytes, symptom triggers, stress, CaMKII

## Abstract

It has been hypothesized since the 1960s that the etiology of schizophrenia is linked to dopamine. In the intervening 60 years, sophisticated brain imaging techniques, genetic/epigenetic advances, and new experimental animal models of schizophrenia have transformed schizophrenia research. The disease is now conceptualized as a heterogeneous neurodevelopmental disorder expressed phenotypically in four symptom domains: positive, negative, cognitive, and affective. The aim of this paper is threefold: (a) to review recent research into schizophrenia etiology, (b) to review papers that elicited subjective evidence from patients as to triggers and repressors of symptoms such as auditory hallucinations or paranoid thoughts, and (c) to address the potential role of dopamine in schizophrenia in general and, in particular, in the fluctuations in schizophrenia symptoms. The review also includes new discoveries in schizophrenia research, pointing to the involvement of both striatal neurons and glia, signaling pathway convergence, and the role of stress. It also addresses potential therapeutic implications. We conclude with the hope that this paper opens up novel avenues of research and new possibilities for treatment.

## 1. Introduction

There are many theories about the etiology of schizophrenia (including dopaminergic [1], glutamatergic [2], inflammatory [3], and developmental [4]), but none deal with the fact that many hallmark manifestations of this disorder fluctuate in their expression from one day to the next, or from one hour to the next. Some symptoms, such as thought intrusion, difficulties understanding language, trouble thinking, trouble distinguishing between fantasy and reality, ideas of reference, and momentary changes in perception, are subtle, and their presence may not be subjectively quantifiable [5]. Other psychotic symptoms, such as auditory hallucinations or surges of paranoid thinking, are more evident to the person and more likely to be remembered and reported. Available mobile technology assessments are now able to capture events, emotions, and stressors that immediately precede the exacerbation or the waning of such symptoms [6,7,8]. We examined evidence for the molecular pathways involved and their role in the association between dopamine (DA) release or altered dopamine receptor (DR) function and these identifiable symptom triggers [9].

This paper reviews the current understanding of the development and etiology of schizophrenia, addresses its DA links and addresses what is currently known about the association of DA with symptom induction and its day-to-day fluctuation.

## 2. Method

The focus of our review’s molecular/biological section was to guide the reader from the development of the first antipsychotics to the current understanding of their antidopaminergic properties and to the presumed role of dopamine in the initiation of pathology.

Our strategy was to build upon a detailed synthesis of available research on schizophrenia pathology (reviewed in [10]) by searching the PubMed database with the following keywords: “Schizophrenia” plus one or more of the following: genetics, prenatal development, prenatal environment, neurodevelopment, monozygotic twins, premorbid pathology, glutamate hypothesis, gamma-aminobutyric acid (GABA), α-amino-3-hydroxy-5-methyl-4-isoxazolepropionic acid (AMPA), N-methyl-D-aspartate (NMDA), neurochemical pathology, dopamine hypothesis, DA circuits, striatum, cognitive, structural, glia, and inflammation. To avoid excessive detail, we focused on the most recent reviews of each topic but also on the most relevant research publications of the last 5–10 years. Classic historical references were included as well.

This completed, in Part 2, we reviewed written narratives by patients and patient responses to direct questioning about symptom triggers by entering the following search terms: “Subjective patient narratives” AND “schizophrenia” into Google Scholar and searching the literature for results of studies on “ecological momentary assessments (EMA)” and “mobile clinical assessment” AND “schizophrenia.” EMA is a data collection method that samples momentary experiences within a person’s natural environment over several days, commonly using specialized smartphone apps. The information is collected in real time so that recall bias is eliminated and temporal connections can be analyzed.

## 3. Molecular Pathways Underlying Schizophrenia Pathology

### 3.1. Dopamine and Schizophrenia

Even before modern antipsychotics, ancient Greeks used an extract from the root of the Rauwolfia plant to alleviate psychosis [11]. The active ingredient was later isolated as reserpine [12].

Advancement in the search for the etiology of schizophrenia finally came with the development of modern antipsychotic drugs. In the 1950s, two French psychiatrists, Delay and Deniker, observed that a newly synthesized drug, chlorpromazine, was effective in reducing positive symptoms of schizophrenia, particularly delusions and hallucinations [13]. Additional phenothiazines and, subsequently, butyrophenone drugs were gradually developed [14,15].

In the 1960s, by studying the mechanisms of action of these drugs, two prominent scientists laid the foundations for our current understanding of schizophrenia etiology. Arvid Carlsson first proposed that the mechanism of therapeutic action of antipsychotics was catecholamine antagonism (reviewed in [16]). This was then extended by Van Rossum, who proposed that antipsychotics were able to reduce symptoms by blocking dopamine receptors, hence dampening the response to DA [17]. This led to the formulation of the “dopamine hypothesis of antipsychotic action.” In the 1970s, this hypothesis was further strengthened by the work of Philip Seeman, who first demonstrated that the dopamine 2 receptor (D2R) was the critical target. His research group showed a strict inverse relationship between the binding affinity for D2R and the mean therapeutic dose prescribed for all antipsychotics in use at the time [18,19].

New, second-generation drugs were developed, which, in addition to targeting the D2R, acted on other signaling pathways and were clinically associated with fewer extrapyramidal side effects, notably tardive dyskinesia [20]. However, the prevailing view continues to be that the blockade of DA in the striatum is a fundamental property of all antipsychotic drugs [21]. This is supported by the observation that second-generation antipsychotics also fall well within the D2R binding affinity vs. effective dosage correlation curve (reviewed in [1]).

It is evident that the dysfunction of the DA system is central to schizophrenia etiology and key to our understanding of schizophrenia symptom fluctuation. However, the cause of its dysregulation remains unknown, although vital clues paint an emerging image of the pathology underlying this complex disease. These clues include genetic predispositions, environmental factors, and altered brain structure and function.

### 3.2. Schizophrenia Is a Result of the Synergistic Interaction between Genetic and Environmental Factors

The heritability of schizophrenia is estimated at 80% [22,23], suggesting that genetic factors play a significant role in the predisposition to the disease. However, studies using identical twins significantly underestimate schizophrenia heritability. Due to unequal sharing of the placenta in a great proportion of identical twin pregnancies, the twins experience an unequal share of nutrients, oxygen, and placental and maternal hormones, but also toxins. As a result, identical twins are more discordant in size and IQ than fraternal twins (re-viewed in [24]). These conditions likely increase schizophrenia susceptibility selectively in the disadvantaged twin only, thus reducing heritability estimates [24].

The most recent and most extensive whole-genome linkage study identified 120 genes associated with schizophrenia [25,26]. For 105 of these, protein expression information is available [27,28,29] and shows that they are expressed in neurons and glial cells (astrocytes, microglia, and oligodendrocytes (ODs)). In particular, 90% are expressed in ODs, with 58% significantly enriched or specific for ODs. However, GWAS-identified genes do not lead to a comprehensive list, as genes with more complex mutations are not detectable by GWAS, such as the CAA insert/TATC deletion in the Nogo/RTN4 gene [10,30,31], which codes for a protein specifically expressed in ODs [32]. (This mutation is as-sociated with schizophrenia in populations where it is prevalent; in populations where it is exceedingly rare, results were inconclusive.) These data indicate that, in addition to the D2R, which was repeatedly identified in GWAS studies, ODs may also play an important role in the neurodevelopmental and/or functional pathology of schizophrenia.

This is also supported by the results of the GWAS performed by the Schizophrenia Working Group of the Psychiatric Genomics Consortium in 2014 [26]. Their results suggest that only a few genes carry a non-synonymous exonic mutation affecting gene function and that most schizophrenia-associated mutations act by altering gene expression [26]. In particular, schizophrenia variants were associated with enhancers active in the brain, in genes expressed primarily in neurons and oligodendrocytes [26]. An extended GWAS study (2022) [25] showed that schizophrenia-associated mutations are mainly expressed in neurons, but also oligodendrocytes, although the latter did not reach statis-tical significance. This may have been due to mutations in glutamatergic and dopaminergic systems being as-signed to neurons rather than to oligodendrocytes [33] and perhaps also due to the limited number of cell types used in the analysis [25].

Though highly heritable, schizophrenia is far from being solely determined by one’s genetic heritage. It results from the synergistic interaction between genetic and environmental factors. In fact, the concordance for schizophrenia among identical twins is 50% [34], illustrating the significance of the role that environmental factors play in the development of the disease even when the genetic predisposition is identical (reviewed in [35]).

The early impact of genetic and environmental fac-tors is thought to lead to prodromal symptoms, which long predate the onset of psychotic symptoms, often by more than a decade. Prodromal symptoms include deficits in social function [36,37], cognitive ability [38], and motor skills. The presence of neurodevelopmental pathology predating the onset of psychosis is supported by magnetic resonance imaging (MRI) and microarray studies showing white matter brain abnormalities [39,40] that are indicative of abnormal early development [41]. The post-pubertal symptomatic onset of psychosis is now considered to be a late disease stage (as re-viewed in [35]).

### 3.3. The Involvement of the Striatum in Schizophrenia Pathology

Analysis of schizophrenia pathology at the function-al level points to the involvement of neural pathways that connect the frontal cortex and the striatum. Functional MRI and positron emission tomography (PET) studies confirm the involvement of the DA system [42,43] and show hyperfunction of the associative striatum (caudate) (reviewed in [44]) and hypofunction of the prefrontal cortex (hypofrontality) [45,46].

While positive symptoms are mediated by hyper-function of the associative striatum [47,48,49], negative symptoms seem to originate from hypofunction of the ventral striatum ([49], reviewed in [50,51]).

However, despite both symptom clusters being D2R-mediated, while antipsychotics are effective in reducing positive symptoms (psychosis), the most debilitating negative symptoms (anhedonia, amotivation, and sociality) and cognitive symptoms (working memory impairment), the onset of which is during the prodromal phase, do not respond to antipsychotic treatment [52,53,54].

Because of the role of the prefrontal cortex (PFC) in cognition, the PFC became the brain focus of a search for the treatment of negative symptoms. The striatum and the PFC are anatomically and functionally linked, with the PFC sending projections to the striatum and the striatum sending feedback to the PFC, thus modulating DA release within the PFC [55], but it was unclear which brain area was the source of the observed pathology. Alterations in several signaling pathways were detected in the PFC, involving the DA, N-methyl-D-aspartate (NMDA), gamma-aminobutyric acid (GABA), and cholinergic systems (reviewed in [2]). This made the NMDAR a potential drug target to treat negative and cognitive symptoms [56,57], but treatments targeted at the NMDA system have shown no efficacy [58,59].

However, DA also plays an important role in PFC-dependent cognitive functions, including working memory, associative learning, cognitive flexibility, and attention [60]. Schizophrenia patients show a decrease in DA release in the cortex [61], which has the potential to mediate the cognitive symptoms of schizophrenia (reviewed in [50]). Importantly, the research group of Christoph Kellendonk has shown that elevation of D2R activity in the striatum is able to replicate the schizophrenia-like pathology of the PFC in experimental animals [62], which suggests that the striatum is the source of the pathology.

Recent research supports the role of DA also in the prodromal stage of schizophrenia, at the onset of negative and cognitive symptoms. An increase in D2Rs is present long before the onset of psychosis, and elevated dopamine synthesis capacity is associated with the se-verity of prodromal symptoms and with the likelihood of conversion to psychosis (reviewed in [63]).

Furthermore, the upregulation of the D2R in the striatum led to the development of positive and cognitive symptom analogs in the mouse model (reviewed in [50]). More importantly, switching off the D2R increase during adolescence prevented the onset of cognitive and social deficits in the mice. This suggests that both negative and cognitive symptoms depend on D2R up-regulation in the striatum early in puberty and that this potentially drives the prodromal stage of the disease (reviewed in detail in [50]). This also supports the hypothesis that early treatment could be effective in pre-venting negative symptoms [64].

### 3.4. Neurodevelopmental Process of the Striatum

Observed pathology, therefore, centers on the func-tion of the striatum. Evidence for the cause of striatal dysfunction converges on a key developmental stage of this brain structure. The striatum undergoes a major developmental change during the second trimester of pregnancy, which is a known susceptibility period when environmental factors increase the likelihood of the later development of schizophrenia [65]. The same developmental process occurs during the second post-natal week in rats [66] and is characterized by the onset of expression of the dopamine 1 receptor (D1R) and D2R, as well as the associated downstream signaling machinery, such as the calcium/calmodulin-dependent protein kinase II (CaMKII) [65]. This is followed by a striking burst in myelination by ODs, coinciding with the appearance of striations [65].

Our understanding of striatal development during this time will help us identify how the environment con-tributes to the development of the disease. For example, iron deficiency affects the expression of transferrin (Tf), which is one of the most highly upregulated genes dur-ing this period [65], and controls the expression levels of several key myelination proteins. Hence, iron deficiency has the potential to significantly alter the prenatal mye-lination process (reviewed in [67]).

Infection during pregnancy is also known to increase susceptibility to schizophrenia in the offspring. Perina-tal inflammation activates microglia and astrocytes and triggers cytokine release, which can result in damage to neurons and ODs [68,69]. Nutritional deficiencies due to starvation, particularly iron deficiency [67], also have the potential to impede the ability of ODs to carry out myelination [70]. This is in line with observations of widespread aberrant myelination in schizophrenia [15,71,72].

The involvement of aberrant myelination is also supported by a key characteristic observation in hu-mans with schizophrenia: ventricular enlargement [73,74] and widespread cortical thinning [75]. This is associated with general atrophy of neurons and reduction in myelination tracts, but not a decrease in neuronal numbers [76,77]. Such pathology is indicative of deficits in neuron-associated ODs ([78], reviewed in [71]). The role of ODs has been historically underestimated, yet the human brain contains approximately the same number of glia as neurons, with about 50% of glia being ODs [79]. Bidirectional communication between neurons and ODs is mediated via OD expression of several neuro-transmitter receptors, including the D2, NMDA, and AMPA receptors (reviewed in [33]), pathways that are also found to play a key role in schizophrenia (re-viewed in [2]).

Understanding how this striatal maturation process is orchestrated may be key to understanding the fundamental causes of D2R dysregulation.

### 3.5. Molecular Pathways Linking Schizophrenia Symptoms and Stress

What we know of the etiology of schizophrenia points to the dysfunction of the DA signaling network, including NMDA, AMPA, and GABA systems. CaMKII is a multifunctional kinase that integrates the D2R-AMPA-NMDA-GABA pathways (reviewed in [10]). It acts as a rheostat, modulating and integrating the activity and response to signals, and transmits these signals to a large number of targets at the synapse [80,81] and in the nucleus [82].

CaMKII is also highly sensitive to the effects of stress [83] and is likely responsible for schizophrenia symptom fluctuations in response to stressful situations. In the cortex, the neuronal CaMKII holoenzyme is com-posed of 12 alpha (α) and beta (β) subunits. CaMKIIβ is the regulatory subunit of the kinase and responds to calcium levels 10× lower than that required for CaMKIIα response [84]; hence, the α:β ratio determines the activity of the kinase.

Prolonged stress results in cortisol release, which in-duces Ca2+ influx and leads to the activation of CaMKII [85], which then regulates D2R expression [82] as well as AMPA and NMDA receptor function [86]. In acute stress, CaMKII mediates the response to noradrenaline [87]. In individuals with increased levels of CaMKIIβ [88,89], this likely results in abnormal reactivity to stress [90] and dysregulation of the glutamatergic [91] and dopaminergic [82] systems (reviewed in [10]).

CaMKII also plays a vital role in neuronal maturation during early postnatal striatal development [92] and during puberty [90], hence making this period susceptible to perturbations that alter CaMKII activity. In rats, stress during the period of striatal development, followed by a second stressor during puberty, results in the upregulation of CaMKIIβ and of the D2R [93].

CaMKII also plays a central role in mediating a key hallmark of schizophrenia pathology observed both in humans and in animal models: enhanced amphetamine-induced dopamine release [1,94,95]. Enhanced amphetamine-induced dopamine release is absolutely dependent on CaMKII activity and has been shown to be inducible by early stress [96]. Furthermore, a key phenotypic marker of animal schizophrenia models, an increase in D2R levels in their high-affinity states (D2high), is also normalized by inhibition of CaMKII [1,97]. The CaMKIIβ subunit was found to be elevated in both humans with schizophrenia and in an animal model, suggesting it plays an important role in schizophrenia pathology in both species [10,88,93,97,98]. Therefore, CaMKII is a point of convergence of several key signaling pathways associated with schizophrenia symptoms and is able to mediate the effects of stress on this signaling network (Figure 1). Inhibition of CaMKII may be an important treatment target, not only to modulate the effects of stress but for schizophrenia in general.

Since CaMKIIβ-specific inhibitors are being developed for the treatment of cardiomyopathy [99], if they are able to cross the blood-brain barrier, it will be important to note their effects in patients who also have schizophrenia.

The next portion of this paper examines how dopa-mine may be involved in the appearance and disappearance of schizophrenia symptoms, as experienced by patients.

## 4. Clinical Manifestations: Symptom Checks and Triggers as Evidenced by Patients

### 4.1. Schizophrenia Symptoms Fluctuate

Schizophrenia symptoms such as hallucinations, for instance, are not always present during the course of a day, but rather, patients report that they come and go. The affective distress induced by psychotic symptoms can also emerge during sleep in the form of nightmares [100]. Many short-term stressors are known to induce transient psychotic symptoms. Since Harry Stack Sullivan first wrote about the influence of the immediate environment on the fluctuation of symptoms in individuals with schizophrenia [101], many personal patient accounts have confirmed it [102].

### 4.2. Social Engagement

Of the many stressors capable of precipitating psychotic symptoms, social stress and interpersonal en-counters are the ones most frequently noted. Marley interviewed schizophrenia inpatients, asking them about the triggers of their two most recent symptom flare-ups [103]. Almost 63% of the 54 participants in his study identified interpersonal interaction as a trigger. Arguments with co-patients were the most frequently cited; slights and criticisms from staff came next. Nearly three-fifths of interviewees also reported that positive inter-personal encounters, such as being on the receiving end of encouragement and support, made symptoms re-cede.

Writing in the personal accounts section of Schizophrenia Bulletin, Colori [104] expressed it this way: “…I’ve realized ……that some of the most stressful situations for me are when I’m with and around people. Generally, ……large social gatherings …causing more hallucinations.”

In 2002, using the experience sampling method, Delespaul and colleagues found that social withdrawal, as well as both relaxation and work activities, resulted in a decrease in hallucinatory intensity, while social engagement and also passive leisure activities, such as watching TV, raised intensity levels [105].

Crosier et al. used Facebook to ask voice-hearing respondents, “What are 3 things that make your voices worse?” Of the 90 persons who responded, 23% of answers involved other people, conflict with others, or, paradoxically, being alone [106]. A smaller percent of responders reported that “Telling others about them [hallucinations]” and “Thinking people have a reason to dislike me” worsened hallucinations.

Bell et al., in a case study using ecological momentary assessment, reported that their subject’s voice in-tensity increased when feeling anxious in the company of several people at once [6]. The intensity decreased when the patient was near someone with whom he felt comfortable.

Social contact has been reported as triggering not only hallucinations but also paranoid thinking, especially whenever the company involved relatively unfamiliar people [107].

Using a structured diary technique that captures mental states and contexts in everyday life, more than one study has shown that fluctuations in experiences of psychosis depend on social contact transitions. Symp-toms are always reported as more frequent in encounters with relative strangers than in contacts with family or friends [108]. A study examining the context of delusions in a clinical population also reported that the presence of familiar individuals decreased the probability of delusions, whereas the risk increased during times of inactivity [109].

The specifics of the social contact, such as subjectively perceived empathy on the part of the other person, are critical to the response. Twenty-nine participants with psychotic symptoms (ages 16–30) in a recent experience sampling study were compared to twenty-eight controls. Empathy was measured using a self-report questionnaire. The findings were that social contact was associated with positive affect, especially when contact was with close others, and that empathy moderated the association between the closeness of contact and positive affect (there was no direct association between empathy alone and positive affect) [110].

Therapist reports have also been examined for clues as to interpersonal processes that trigger symptoms. A study by Leonhardt et al. (2018) searched for factors in conversations between patients and therapists that were able to predict subsequent psychotic content [111]. For-ty-eight transcripts from one patient’s experience identified passages of delusional or disorganized content and noted the conversational themes that had immediately preceded these passages. The therapist setting boundaries in the therapeutic relationship or the therapist challenging the patient’s interpretation of events appeared to trigger psychotic content, especially when this occurred early in the treatment process when the alliance was not yet well established.

Social situations can lead to positive and negative affect, and, past a threshold level, either one can result in symptoms. It is also possible that stressful social situations lead directly to symptoms, which then color affect. A positive or negative emotional state is important, as it is a major determinant of whether social occasions are sought out or avoided. Using ecological momentary assessments, emotional states and motivation to socialize were studied for 7 days in 105 patients with schizophrenia and 76 non-clinical controls. In both patients and controls, positive emotions created an incentive to socialize, and negative emotions dampened that incentive. The controls, however, recovered from negative emotions much more quickly than patients [112]. Why it takes schizophrenia patients longer than usual to re-cover from negative emotions is not well understood. A questionnaire was administered to 71 patients with psychosis and 42 controls in the hope of determining which of several possible strategies (reappraisal, acceptance, awareness, suppression, rumination, distraction, social sharing) were used to recover from negative emotions, but the study was unable to reach clear conclusions about group differences [113].

Though time away from socializing (downtime, solitude, and self-motivated isolation) has been reported as beneficial in schizophrenia [114,115], some psychotic symptoms, e.g., paranoia, can be aggravated by isolation. Fett et al. (2022) studied 29 patients with psycho-sis, 20 first-degree relatives, and 26 controls using the experience sampling method for one week, with 10 samplings daily [116]. In contrast to both relatives and controls, patients in this study experienced significantly greater paranoia when alone than when in company, and the more familiar the company, the less the result-ant paranoia.

It is probable that being alone, while sometimes sought out and sensed as positive, can also be experienced as a consequence of social exclusion and, thus, become a breeding ground for psychotic symptoms [117]. Much would depend on whether solitude was perceived as chosen or involuntarily imposed.

### 4.3. Role of Dopamine in Social Affiliation

When individuals experience something pleasing (such as the taste of sugar, for instance), the reward systems of the brain, located along dopamine pathways, are activated. Whenever one encounters something re-warding, the brain responds by releasing more dopa-mine, mostly from the ventral tegmental area (VTA) (ventral striatum), in the midbrain. From there, dopa-mine is transported throughout the brain along different routes, the two main ones being the mesolimbic and mesocortical pathways. Because dopamine is associated with reward [118], its signaling increases the motivation to repeat the pleasing activity or deliberately seek out the circumstances that led to the pleasure. Dopamine coats these activities or circumstances with special im-portance. This phenomenon has been studied in voles, animals that show strong social bonding. Goodwin et al. (2019) found that centrally administered D2R agonists promote partner preference, whereas D2R antagonists prevent partner formation [119]. In terms of peer relationships, the DR agonist apomorphine facilitates peer preferences in voles, while the haloperidol blockade of DRs, no matter the dose used, does not alter peer preferences. These results suggest that prairie vole peer relationships, while rewarding, are less dependent on dopamine signaling than are intimate partner preferences. Inhibition of dopamine neurons of the VTA in mice and rats has been shown to reduce their approach-es to unfamiliar members of their species [120,121]. In fish that breed cooperatively, Antunes et al. (2022) [122] demonstrated that the D1-like and D2-like receptor pathways are both involved, but differently, in the modulation of affiliative behaviors.

One research team was successful in studying hu-man bonding (between mother and child) [123] and demonstrated that human maternal bonding was significantly associated with striatal dopamine function and dependent on reward circuits in the brain. The re-search team used a combined functional MRI-PET scanner to examine dopamine responses in mothers as they gazed at their infants and subsequently analyzed the connections between the nucleus accumbens, the amygdala, and the medial PFC, the “medial amygdala network” that supports social functioning. The results suggested that a mother’s response to her infant was associated with increased dopamine secretion and in-creased strength of the connections within the network. Using positron emission tomography, it has also been shown that patients with social anxiety disorder have more D2R availability in the bilateral orbitofrontal cortex and right dorsolateral prefrontal cortex than do healthy controls [124,125] and that this extra availability disappears when social anxiety symptoms wane.

Despite few and inconsistent results, various alterations in dopamine transmission have been found in patients who are socially avoidant [126]. There is a reduced density of dopamine uptake sites and a low D2R binding potential in the ventral striatum. When symptoms improve, D2R binding increases in medial pre-frontal and hippocampal regions, and prefrontal D2R density rises. The role of dopaminergic signaling in social avoidance appears to be highly dependent on its localization within the circuitry of the brain.

### 4.4. Motor Activity

One potential way schizophrenia patients have found to improve their social comfort is by becoming more active because exercise is associated with im-proved social skills [127], while sedentary behavior ap-pears to increase the frequency of delusions [109]. Kimhy et al. (2015) compared an aerobic exercise program group with treatment as usual over a 12-week time period [128]. The study results showed that aerobic exercise improved social function, as judged by both patients and their caregivers. A similar effect of activity had already been shown by Chamove in 1986 [129] and has been shown since [130].

This effect may be mediated by a positive effect of motor activity on mood [131] and by a temporary de-crease in anhedonia, one of the negative symptoms of schizophrenia [132].

### 4.5. Dopamine and Exercise

The improvement of many mental health conditions after exercise has been attributed to observed changes in brain connections that depend on catecholamine path-ways. This agrees with indirect evidence that improvement after exercise is mostly seen in executive functions, the functions most impacted by dopaminergic disruption [133].

Exercise has variously been shown to improve positive and negative symptoms, quality of life, cognition, and hippocampal plasticity and also to increase hippocampal volume in the brains of patients with schizophrenia [134]. Though studies regarding adaptations in the dopaminergic system in response to exercise in schizophrenia patients remain sparse, there is evidence that the increase in serum calcium resulting from exercise enhances brain dopamine synthesis in a region-specific manner. Dopamine levels rise in the hypothalamus and midbrain after aerobic exercise, but they drop in the PFC, hippocampus, and striatum [135]. Exactly how dopamine mediates the beneficial effects of exercise on symptoms of schizophrenia remains unknown, but it is assumed that the effects are indirect, mediated perhaps through a reduction in anhedonia [136].

### 4.6. Effects of Sleep

Fragmented sleep, sleep hallucinations, night anxiety, insomnia, and parasomnias have been positively correlated with psychotic-like experiences [137]. In help-seeking university students, sleep quality moderates the relation between psychotic-like experiences and suicidal ideation. Better sleep appears to be protective [138]. After controlling for sociodemographic variables, depression, and drug/alcohol abuse, poor sleep quality made a unique contribution in one study (accounting for 5.8% of the variance) to the level of distress occasioned by attenuated psychotic symptoms [139]. In an online sur-vey, however, poor sleep as a precursor to worsening symptoms in persons with psychotic illness could not be demonstrated [140].

### 4.7. Effects of Distress

Inability to sleep is commonly associated with worries related to distress. There have, however, been few experimental studies relating the experience of distress to the worsening of specific symptoms in patients with schizophrenia.

Kimhy et al. (2017) studied arousal by investigating the link between cardiac autonomic regulation and the appearance or exacerbation of auditory hallucinations [141]. This was a 36 h experience sampling study of 40 schizophrenia patients carrying mobile electronic devices that allowed them to report on the severity of auditory hallucinations ten times a day. Patients wore a Holter monitor that monitored their electrocardiograms continuously during that time. Power spectral analysis was used to determine heart rate variability, a measure of autonomic arousal, during the 5 min prior to each experience sample. The results were a small but significant inverse correlation between parasympathetic input and the coming and going of auditory hallucinations.

Using a repeated-measures design, 64 non-clinical participants who experienced occasional psychotic symptoms were randomly assigned to a noise stress or non-stress condition. Under these conditions, both groups were asked difficult knowledge questions. An increase in paranoia, depression, and negative emotion was seen in the stress-exposed group. The degree of paranoia induced by stress was moderated by the level of a person’s pre-existing vulnerability to paranoia and was mediated by the extent of anxiety. In other words, anxiety was most strongly associated with paranoia in those who had, at baseline, been assessed as most vulnerable [142]. In this experiment, the noise was not un-der the participants’ control. It would have been interesting if a subgroup had been able to shut off the noise because dopamine mediates control-mediated process-es. Using [11C] raclopride PET, Vassens et al. (2019) test-ed the relationship between D2R binding in the striatum and locus of control (LOC) in 15 healthy volunteers [143]. The results showed a large positive correlation between increased striatal D2 binding and external LOC, which is the tendency or cognitive bias to perceive environmental events as largely uncontrollable.

### 4.8. Dopamine in Sleep

Sleep problems and circadian rhythm disruption are commonly found in schizophrenia and are associated with symptom severity. There is substantial documentation on dopamine’s involvement in the regulation of the sleep/wake cycle, in which it acts to promote wake-fulness: elevated dopamine levels can impair sleep. There is also evidence for the influence of dopamine on the circadian clock through entrainment of the master clock in the suprachiasmatic nuclei (SCN). Dopamine signaling itself is under circadian control. It appears that dopamine and the circadian system that controls sleep relate bidirectionally to schizophrenia symptoms, such that disturbances to one exacerbate abnormalities in the other [144].

Dopamine is involved in regulating sleep and wake-fulness across all species [145]. Generally, it promotes arousal; its effects, however, depend on the brain region and receptor subtypes involved. Dopamine signaling is involved in the functioning of two key components of the circadian system: The retina and the suprachiasmatic nucleus of the hypothalamus. There is also evidence for dopamine clock regulation through inhibition of prolactin in the hypothalamus and melatonin in the pineal gland. A hyperdopaminergic mouse model of schizophrenia shows aberrant ultradian activity rhythms, mirroring day/night disruptions seen in schizophrenia patients [146]. Many patients with schizophrenia sleep during the day and are up, instead, at night, though this has also been attributed to a motivated avoidance of social contact [147].

It is difficult to study the effects of stress on dopa-mine in schizophrenia patients because patients with this diagnosis are almost always in treatment with antipsychotic drugs that block dopamine receptors (DRs). One strategy has been to recruit symptom-free first-degree family members of schizophrenia patients who, it is assumed, share a genetic vulnerability to the dis-ease. Van Leeuwen et al. (2019) investigated reward processing in response to acute stress in healthy con-trols and siblings of schizophrenia patients [148]. In healthy controls, the investigators found the expected increased ventral striatum and orbitofrontal cortex response to positive feedback following stress, but this was not present in siblings of schizophrenia patients. Siblings of schizophrenia patients are also reported to show unpredictability in the dopamine response to stress [149].

To study the putative association between dopamine (DA) hyperreactivity and psychotic reactivity to stress, MyinGermeys et al. (2005) recruited nonmedicated first-degree relatives of psychotic patients and general population controls [9]. Altered DA reactivity was de-fined as elevations of plasma homovanillic acid (HVA), a dopamine metabolite. All trial participants received a digital wristwatch and completed assessment forms 10 times a day on 6 consecutive days, prompted by a beep from the watch. At each beep, they reported current thoughts, symptoms, and mood. The stressor was an in-fusion of a glucose analog that produces transient hypoglycemia. The placebo control was an infusion of iso-tonic saline. The final study sample consisted of 96 subjects. Psychotic symptom intensity was significantly higher in the psychosis relatives than in the controls. No significant differences were found in HVA reactivity. Because of this, it has been suggested that, rather than increased DA secretion, elevated D2R density could be causing DA sensitization [43] with increased transmission via the D2R [150].

### 4.9. Substance Use

Stimulants such as amphetamines are dopaminergic and have been known to induce psychotic symptoms in healthy individuals [42]. When individuals with schizophrenia take such stimulants, psychotic symptoms are aggravated [151]. Patients with psychosis are also extra sensitive to the effects of alcohol. A small amount of alcohol impairs cognition and produces transient increases in positive psychotic symptoms and perceptual al-terations but does not affect negative symptoms [152].

Henquet et al. (2010), using a structured time-sampling technique, studied the effect of cannabis in 42 patients with psychosis and 38 control subjects [153]. Cannabis increased the frequency of hallucinatory experiences but also improved positive affect and decreased negative affect. As a result of reports collected via four in-person interviews with two general population cohorts over a ten-year period, van Os et al. (2021) found that the use of cannabis correlated positively with sub-sequent psychotic experience [154]. Investigating whether drugs of abuse exacerbate or ameliorate schizophrenia symptoms, Menne and Chesworth (2020) re-viewed the preclinical literature on rodent models of schizophrenia and concluded that these drugs can do both, and that the effect depends on the model used and, to a large extent, on the specifics of the drug [155]. Nicotine was ameliorative in most models, which may help to explain the high rate of smoking seen in schizophrenia [156].

### 4.10. Dopamine and Substance Use

Many drugs of abuse stimulate dopamine release in the ventral striatum and PFC, a fact that has been con-firmed through imaging studies [154] and should theoretically increase schizophrenia symptoms [155]. Proebstl et al. (2019) reviewed the effects of stimulant use on DA function using PET and SPECT imaging [156]. They examined 39 studies on 655 cocaine, amphetamine, methamphetamine, or nicotine users and 690 healthy controls. They looked separately at D2/D3 receptors and dopamine transporters (DAT) in the stria-tum (caudate and putamen) and found so many contradictory results that a meta-analysis of nicotine use could not distinguish effects on smokers from those of non-smokers. With respect to cocaine, users showed a significant decrease in DR availability vis à vis controls in all of the examined regions, while striatal DAT avail-ability was significantly increased. In methamphetamine users, the density of DR and DAT was decreased in all regions, and DAT availability was down in all regions compared to controls. Lowered DAT availability was also true for amphetamine users. In other words, for cocaine, methamphetamine, and amphetamine, there was a general down-regulation of DA, which is a surprising result.

## 5. Discussion

The etiology of schizophrenia begins with genetic variants that impart sensitivity to a variety of environ-mental factors, particularly during the fifth month of gestation in humans, when the striatum is actively maturing [65]. Exposure to such factors then leads to brain changes. Given that many schizophrenia-associated genetic variants occur in genes expressed in ODs [25], this process likely involves OD function and abnormal myelination [15,68,69], perhaps underlying later neuronal dysfunction. This is also supported by the presence of white matter abnormalities [39,40]. However, the mechanism of how this impact occurs is still not clear, although it invariably leads to a characteristic alteration in the D2R signaling system in the striatum, a key aspect of schizophrenia [1,44].

Alterations in other pathways have also been detect-ed, particularly in the NMDA, AMPA, and GABA systems (reviewed in [2]). Importantly, NMDARs, AM-PARs, GABARs, and D2Rs all converge on CaMKII for downstream signal transduction and regulation (re-viewed in [10]). The beta (regulatory) subunit of CaMKII was found to be upregulated in schizophrenia, both in humans and in an animal model, indicating that it plays an essential role in the disease, likely by increasing the enzyme’s sensitivity to calcium signals [88,89,93,156]. CaMKII also controls several key phenotypic aspects of schizophrenia, such as enhanced amphetamine-induced dopamine release [42,94,95,98] and elevated D2High binding states [156]. Hence, CaMKIIβ forms a central point of convergence of schizophrenia pathology (Figure 1). Interestingly, CaMKIIβ is also the predominant CaMKII subunit expressed in oligodendrocytes, where it regulates their maturation and CNS myelination [157], as well as oligodendrocyte–neuron communication via NMDA, AMPA, and D2 receptors [33].

The disease progresses in stages: a neurodevelopmental stage when alterations in function impart pre-disposition to the disease and a prodromal stage when D2R alterations seem to cause ongoing deterioration concurrent with the onset of negative and cognitive symptoms, which may be ameliorated by early treatment. The last stage is the development of psychosis and the irreversibility of negative and cognitive symptoms by antipsychotic treatment.

However, even at the advanced stage, patient re-ports show that schizophrenia symptoms are not constant and fluctuate in response to environmental stimuli, pointing to possible therapeutic strategies. The triggers are invariably related to the experience of stress. In general, stress reduction alleviates schizophrenia symptoms, while stressful situations worsen them. This is as-sociated with the function of DA in mediating the effects of stress and pleasure, and strategies based on this premise may provide an important augmentation to treatment for effective treatment.

Merging clinical observations with molecular evidence indicates that CaMKII may play an important role in modulating symptom severity, as it is very sensitive to stress and conveys its effects to the signaling pathways it regulates, including the D2R and glutamatergic pathways (Figure 1) [83]. Therefore, in addition to stress reduction, pharmacological inhibition of CaMKIIβ may provide a novel therapeutic approach.

## 6. Conclusions

We have approached this narrative review by presenting the dopamine system as a key aspect of schizophrenia pathology, but we have also broadened our horizons by incorporating emerging evidence to present a comprehensive view. Most reviews focus on what causes schizophrenia symptoms to emerge, on the assumption that once emerged, they persist indefinitely unless kept under wraps by effective treatment. In reality, many symptoms come and go and return whether the person is under treatment or not. This review, therefore, focuses on events and circumstances that appear to bring on or dissolve characteristic symptoms and asks whether such symptomatic fluctuations are connected to dopamine. It is our hope that this review will interest readers and will encourage research exploration and the development of new treatment possibilities.

## Figures and Tables

**Figure 1 healthcare-10-01713-f001:**
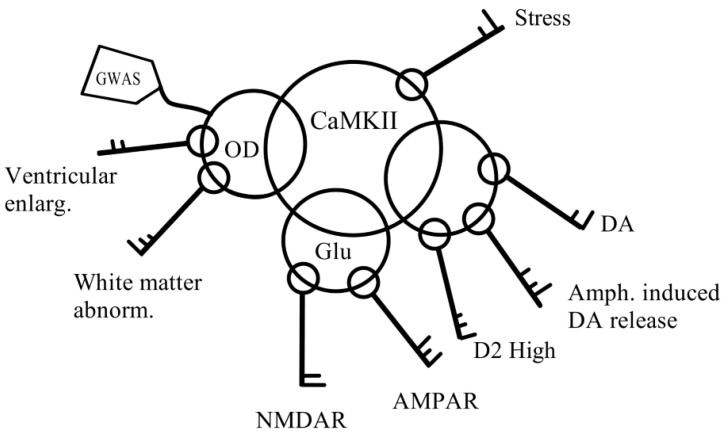
CAMKII integrates pathways that mediate fluctuating effects of stress on psychotic symptoms. Abnorm (abnormalities); AMPAR (α-amino-3-hydroxy-5-methyl-4-isoxazolepropionic acid receptor); Amph. (amphetamine); CaMKII (calcium/calmodulin-dependent protein kinase II); DA (dopamine); D2High (high-affinity state of the dopamine 2 receptor); D2R (dopamine 2 receptor); GWAS (genome-wide association studies); NMDAR (N-methyl-D-aspartate receptor); OD (oligodendrocytes).

## Data Availability

Not applicable.

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
