# Peer review of "Dopamine, Psychosis, and Symptom Fluctuation: A Narrative Review"

_healthcare, 2022, doi:10.3390/healthcare10091713_

Round 1

Reviewer 1 Report

This is a very interesting review. However, this work is very poorly organized. Authors should reconsider the order of presentation of the material. It is better not to separate the parts dedicated to the same problem: for example, stress-schizophrenia and stress-dopamine. The authors need to formulate the aim of the review and the main conclusions more clearly. I would advise adding diagrams or figures to illustrate the main idea.

Author Response

We would like to thank the reviewer for very constructive feedback, which helped us to significantly improve our manuscript. We hope that we were able to address all the recommendations.

Authors should reconsider the order of presentation of the material. It is better not to separate the parts dedicated to the same problem: for example, stress-schizophrenia and stress-dopamine. The authors need to formulate the aim of the review and the main conclusions more clearly.

Response: We have more clearly outlined the focus of the individual sections through revisions of the introduction and by dividing the main text into two clear sections: molecular and patient narratives. These are then merged in the Discussion section, where we highlight the overlap between the molecular aspects and clinical observations, and extend this to spotlight a possible new treatment avenue.

  1. I would advise adding diagrams or figures to illustrate the main idea.

Response: We have added Figure 1 to illustrate the main idea.

Reviewer 2 Report

The authors conducted a review focused on etiology of schizophrenia, triggers and repressors of symptoms as well as the role of dopamine. The review is interesting and well written. I believe it provides a relevant contribution to the field since the point of view is quite different compared to other reviews focused on similar topics. I only have minor comments:

- While extremely brief, I think the introduction provides a good framework of the main aims of this article. However, it might benefit of a (small) expansion, especially in the first lines in which the authors state that there are many hypothesis about the etiology of schizophrenia, without mentioning briefly these hypotheses and relative references. 

- I think the authors should explicitely report that this is a narrative review

- When discussing genetic variants associated with schizophrenia, the GWAS of the Psychiatric Genomics Consortium should be mentioned and discussed (see PMID: 25056061 and PMID: 35396580 for instance). 

- Formatting should be revised as some parts present bold letters or sentences that seem to be marked with another color

- Summarizing some of the most relevant results among those reviewed with a figure, or a table, could be of help for readers

Author Response

While extremelv brief. I think the introduction provides a good framework of the main aims of this article. However, it might benefit of a (small) expansion,especiallv in the first lines in which the authors state that there are manv hypothesis about the etiology of schizophrenia, without mentioning briefly these hypotheses and relative references.

Response: We have added the requested information to the Introduction section and clarified related relevant points throughout the manuscript.

I think the authors should explicitly report that this is a narrative review

Response: This is now in the Title and in the Conclusion.

When discussing genetic variants associated with schizophrenia, the GWAS of the Psvchiatric Genomics Consortium should be mentioned and discussed (see PMID: 25056061 and MID: 35396580 for instance).

Response: Discussion of both publications has been added to the text (lines 131 to 141). Thank you!

Formatting should be revised as some parts present bold letters or sontences that seem to be marked with another color

Response: We're not sure how that happened. It should be fixed now. Revisions are in yellow.

 Summarizing some of the most relevant results among those reviewed with a figure. or a table. could be of help for readers

Response: Thank you. Both reviewers made this recommendation and we have now included a summary Figure based on the integration of available evidence. 

Round 2

Reviewer 1 Report

The manuscript hasThe manuscript has become much better. The text mentions Figure 1, but I didn't find it. I'd like to see this figure. It should summarize the review data.

Author Response

We sent the Figure by url to the journal because we had trouble send it by tiff. Perhaps the journal had similar trouble send it to reviewers, Here is the erl

https://public.3.basecamp.com/p/kD15Q6LFvsKBXEwHemTCvh59

Here is the Legend:

Legend for Figure 1.

Abnorm. (abnormalities)

AMPAR (α-amino-3-hydroxy-5-methyl-4-isoxazolepropionic acid receptor)

Amph. (amphetamine)

CaMKII (calcium/calmodulin-dependent protein kinase II)

DA (dopamine)

D2High (High affinity state of the dopamine 2 receptor)

D2R (dopamine 2 receptor)

GWAS  (genome-wide association studies)

NMDAR (N-methyl-D-aspartate receptor)

OD (oligodendrocytes)

CAMKII integrates pathways that mediate fluctuating effects of stress on psychotic symptoms